# Neonatal Mucormycosis: A Rare but Highly Lethal Fungal Infection in Term and Preterm Newborns—A 20-Year Systematic Review

**DOI:** 10.3390/tropicalmed10040086

**Published:** 2025-03-24

**Authors:** Alfredo Valdez-Martinez, Mónica Ingrid Santoyo-Alejandre, Roberto Arenas, Claudia Erika Fuentes-Venado, Tito Ramírez-Lozada, Fernando Bastida-González, Claudia Camelia Calzada-Mendoza, Erick Martínez-Herrera, Rodolfo Pinto-Almazán

**Affiliations:** 1Sección de Micología, Hospital General “Dr. Manuel Gea González”, Tlalpan, Mexico City 14080, Mexico; alfredovaldezmart@gmail.com (A.V.-M.); rarenas98@hotmail.com (R.A.); 2Hospital General del Zona 33, Instituto Mexicano del Seguro Social, Nayarit 63735, Mexico; ingrid.santoyo.med@gmail.com; 3Sección de Estudios de Posgrado e Investigación, Escuela Superior de Medicina, Instituto Politécnico Nacional, Plan de San Luis y Díaz Mirón, Mexico City 11340, Mexico; cefvenado@hotmail.com (C.E.F.-V.); cccalzadam@yahoo.com.mx (C.C.C.-M.); 4Servicio de Medicina Física y Rehabilitación, Hospital General de Zona No 197, Texcoco 56160, Mexico; 5Servicio de Ginecología y Obstetricia, Hospital Regional de Alta Especialidad de Ixtapaluca, Instituto Mexicano de Seguro Social para el Bienestar (IMSS-BIENESTAR), Carretera Federal México-Puebla Km 34.5, Ixtapaluca 56530, Mexico; titolozada@yahoo.com.mx; 6Laboratorio de Biología Molecular, Laboratorio Estatal de Salud Pública del Estado de México, Toluca de Lerdo 50130, Mexico; mijomeil@hotmail.com; 7Fundación Vithas, Grupo Hospitalario Vithas, 28043 Madrid, Spain; 8Efficiency, Quality, and Costs in Health Services Research Group (EFISALUD), Galicia Sur Health Research Institute (IISGS), Servizo Galego de Saúde-Universidade de Vigo (UVIGO), 36213 Vigo, Spain

**Keywords:** mucormycosis, neonate, gastrointestinal mucormycosis, cutaneous mucormycosis, rhino-orbito-cerebral mucormycosis, fungal infection

## Abstract

Background/Objectives: Mucormycosis is a rare but life-threatening fungal infection, particularly in neonates, due to their undeveloped immune system. This systematic review aims to analyze the risk factors, clinical presentations, treatments, and outcomes of neonatal mucormycosis reported between 2004 and 2024. Methods: A systematic literature search was conducted in PubMed, Scopus, and Web of Science following PRISMA guidelines. Only studies reporting cases of mucormycosis in neonates (≤28 days old) were included. Data on risk factors, clinical features, diagnostic methods, antifungal therapies, surgical interventions, and outcomes were extracted and analyzed. Results: A total of 44 studies met the inclusion criteria, comprising 61 neonatal cases. The most common clinical presentations were gastrointestinal (n = 39), cutaneous (n = 19), rhino-orbito-cerebral (n = 2), and disseminated mucormycosis (n = 1). Diagnosis was primarily based on histopathology (93.4%) and fungal culture (26.2%). The main antifungal treatment was liposomal amphotericin B (63.9%), often combined with surgical debridement (60.6%). Mortality rates remained high (47.5%), particularly in cases of prematurely extreme neonates with angioinvasive disease or delayed diagnosis. Conclusions: Neonatal mucormycosis remains a severe condition with high morbidity and mortality. Early diagnosis through a combination of clinical suspicion and laboratory confirmation, along with prompt antifungal therapy and surgical management, apparently is crucial for improving outcomes. Further studies are needed to optimize treatment strategies and improve neonatal survival.

## 1. Introduction

Formerly classified as zygomycetes, Mucorales have been reclassified to include the pathogenic genera *Rhizopus*, *Mucor*, *Lichtheimia* (formerly *Absidia*), *Saksenaea*, *Rhizomucor*, *Apophysomyces*, and *Cunninghamella* [1]. Among these, the most studied species are *Rhizopus*, *Mucor*, and *Lichtheimia* [1,2,3] (Figure 1). 

These fungi, ubiquitous in nature, exhibit broad, nonseptate, and coenocytic hyphae, causing opportunistic infections in both pediatric and adult populations, particularly those with some form of immunocompromise [1,4]. This infection, known as mucormycosis, is characterized as rare but highly invasive and potentially fatal [5]. Neonates or newborns, due to their undeveloped immune system or prematurity-related factors, represent a particularly susceptible group [5,6,7]. In newborns, mucormycosis exhibits distinctive clinical characteristics depending on the affected topography. The most reported forms are gastrointestinal, which carries a high mortality rate and is associated with enteral feeding and tissue hypoxia [1,3,5,7]; cutaneous, resulting from traumatic lesions or invasive procedures that allow fungal entry [3,5]; and the less reported one, rhinocerebral, acquired through the inhalation of spores from the environment [5].

A crucial aspect of this review is the distinction between preterm and term neonates. All newborns exhibit some level of immune system immaturity [5,6]; however, this immaturity is more pronounced in preterm infants due to deficits in cytokine production and decreased phagocytic activity of neutrophils. While term neonates are generally more immunocompetent, external factors or perinatal conditions, such as the empirical use of broad-spectrum antibiotics and conditions with low oxygen availability, can still predispose them to mucormycosis [1,2,6,8]. Despite advancements in neonatal care, mucormycosis remains a challenging entity to diagnose and treat, with high morbidity and mortality rates in this group [5].

Although there are few studies that have systematically investigated this condition in both children and adults, the available evidence is limited and only partially informs current treatment recommendations [4,5,9]. This highlights the necessity for a comprehensive analysis in more vulnerable populations, like neonates. A deeper understanding of the epidemiological and clinical distinctions between term and preterm neonates, as well as the unique characteristics of different topographic presentations, is crucial for guiding early diagnosis and enhancing the management of these cases.

## 2. Materials and Methods

This systematic review was conducted based on the PRISMA (Preferred Reporting Items for Systematic Reviews and Meta-analyses 2020) guidelines [10] (Figure 2). Three databases, PubMed, Scopus, and Scielo, were searched for Adaptive Clinical Trial, Case Reports, Classical Article, Clinical Study, Clinical Trials, Comparative Study, Editorial, Evaluation Study, Letter, Multicenter Study, Observational Study, and Twin Study. Search MESH terms included “Mucormycosis” OR “Zygomycosis” AND “Neonates” OR “Newborn”. We use limitations on PubMed and Scopus such as less than 1 month of age and studies that include humans. We limited the review to articles in English or Spanish articles and those that were published from 2004 to 2025. Two independent reviewers, Valdez and Santoyo, evaluated the titles, abstracts, and full texts of each potential study. Arenas, Pinto-Almazán, and Martínez-Herrera worked out the details regarding the inclusion to the study. To ensure data accuracy, duplicate studies were excluded. Data were collected just from articles that involved at least 7 out of the following 10 variables:1.Mucormycosis as an etiologic agent.2.Documentation of infection: Mucormycosis was confirmed either histologically by culture or by mycological study (pre- or post-mortem).3.Onset age: Less than 1 month.4.Gestational weeks: Specifying whether the newborn was preterm or term.5.Birth weight.6.Sex.7.Topography of the mucormycosis lesions.8.Therapeutic intervention: Use of antifungal therapy.9.Surgical intervention.10.Outcome: Mortality was assessed as ‘‘all-cause mortality’’ during the course of mucormycosis.

After this initial series of reports was reviewed, the references cited in the above articles were screened for additional cases of neonatal mucormycosis. Their references were carefully scrutinized for single case reports or case series, and those that met the criteria were included.

**Figure 2 tropicalmed-10-00086-f002:**
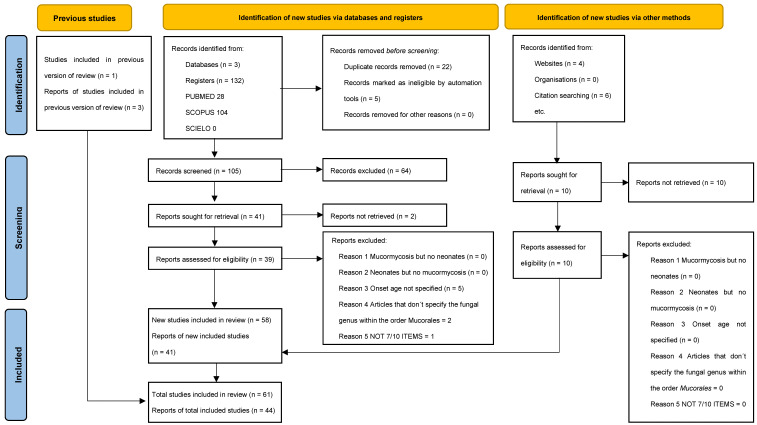
Prisma 2020 flowchart of the data extracted for the systematic review.

### 2.1. Quality Assessment

To assess the quality of the included cases, we utilized the critical appraisal Checklist for Case Reports offered by the Joanna Briggs Institute. We reviewed the inclusion and exclusion criteria, descriptions of clinical history, clinical presentations, interventions, and outcomes. The methodological quality of each article was independently evaluated by two reviewers (Valdez-Martinez and Santoyo-Alejandre). To assess the quality of our systematic review, we applied the PRISMA 2020. A meta-analysis was not feasible due to the lack of quantitative data in the case reports analyzed.

### 2.2. Statistical Evaluation

A descriptive statistical analysis was carried out in order to obtain a detailed understanding of quantitative and qualitative variables of this study. The chi-square statistical test was applied to evaluate the association between variables. A level of evidence of *p* < 0.05 was interpreted as a significant relationship. The chi-square test was used in the univariate analysis. Independent determinants of mortality were determined by multivariate logistic regression analysis. A *p* value of ≤0.05 was considered statistically significant. To carry out this analysis, the statistical package Prism 9.0 (GraphPad, San Diego, CA, USA) was used.

## 3. Results

During the article search, a previous review was found that included three studies published within the period covered by this review, which were therefore included. A total of 61 cases of neonatal mucormycosis were identified from 44 publications. These 44 publications (Table 1 and Table 2) comprised 18 term neonates and 43 preterm neonates [11,12,13,14,15,16,17,18,19,20,21,22,23,24,25,26,27,28,29,30,31,32,33,34,35,36,37,38,39,40,41,42,43,44,45,46,47,48,49,50,51,52,53,54].

Among the preterm neonates (n = 43), many cases were reported in India (n = 15), followed by the USA (n = 9), the Czech Republic (n = 6), Japan (n = 3), and the United Kingdom (n = 2) and one case each in Austria, China, Colombia, Lithuania, Mexico, Saudi Arabia, Türkiye, and Venezuela. The group included 15 females, 23 males, and 5 cases with unspecified sex. Gastrointestinal mucormycosis was the most prevalent presentation with 25 cases, followed by 17 cutaneous and 1 disseminated infection (Figure 3).

All neonates in this group were preterm, with an average gestational age of 28.8 weeks. The average age of onset was 8 days for males, 10.4 days for females, and 6.2 days for cases with unspecified sex. Birth weights varied, with females ranging from 0.46 kg to 2.94 kg (average: 1.21 kg), males from 0.45 kg to 2.1 kg (average: 1.13 kg), and cases with unspecified birth sex from 0.45 kg to 1.4 kg (average: 0.82 kg).

Among these neonates, 29 reported respiratory distress of whom 93.1% (n = 27) received supplemental oxygen therapy.

Prior antibiotic treatment was administered to 67.4% (n = 29) of preterm neonates. In 9 cases, the duration of antibiotic therapy was documented, with an average treatment duration of 13.44 days. Glycopeptides and beta-lactams were the most used antibiotic families, followed by aminoglycosides (Figure 4).

Antifungal treatment was administered in 76% (n = 11) of preterm neonates. Amphotericin B was the most commonly used antifungal drug (Figure 5).

The most frequent sites of angioinvasion in preterm neonates were gastrointestinal (n = 22), cutaneous (n = 7), and disseminated (n = 1). Antifungal therapy was administered to 72% (n = 31) of these neonates, with a median treatment duration of 39.7 days. For those who received only antifungal therapy (n = 8), the mortality rate was 37.5% (n = 3). For cutaneous infections treated with antifungal therapy alone (n = 7), the mortality rate was 28.5% (n = 2). No cases of gastrointestinal involvement received antifungal treatment alone.

Surgical treatment was performed on 31 neonates. Eight cases with gastrointestinal involvement underwent surgical treatment alone, resulting in a mortality rate of 75% (n = 6). Combined antifungal and surgical treatment were administered to 53.4% (n = 23) of neonates, with a mortality rate of 34.7% (n = 8). Additionally, 4 cases (1 cutaneous and 3 gastrointestinal) received neither surgical nor antifungal treatment, with a mortality rate of 100%.

The overall mortality rate among preterm neonates with mucormycosis was 51.2% (n = 22). Mortality rates were 53.3% (n = 8) among females and 43.4% (n = 10) for males.

Mortality rates in preterm neonates varied by infection site: 29.4% (n = 5) for cutaneous mucormycosis and 64% (n = 16) for gastrointestinal mucormycosis. When categorized by prematurity, for extremely preterm neonates, the mortality rate was 61.9% (n = 13); for very preterm neonates, it was 50% (n = 3); for preterm neonates, the mortality rate was 16.6% (n = 2); while for those with unknown gestational age, the mortality rate was 100% (n = 4).

Among the 18 term neonates, the majority (n = 14) were reported in India, with one case each in Austria, Germany, Oman, and Pakistan. These included 4 females, 13 males, and 1 individual with unspecified sex. Gastrointestinal mucormycosis was the most common presentation, accounting for 14 cases, followed by cutaneous and rhino-orbito-cerebral mucormycosis with 2 cases each. Clinically, most neonates were born at term, with an average age of onset of 9.6 days for males, 10.25 days for females, and 7 days for cases with unspecified sex. Birth weights ranged from 1.95 kg to 2 kg (average: 1.97 kg) in females and from 1.7 kg to 2.75 kg (average: 2.2 kg) in males.

Respiratory distress was reported in 11 cases, with 81.8% (n = 9) of these neonates requiring supplemental oxygen.

Additionally, 66.6% (n = 12) of term neonates received prior antibiotic treatment, although the duration of therapy was documented in only 3 cases, averaging 4.6 days. Unspecified or broad-spectrum antibiotics were the most prescribed, followed by glycopeptides and beta-lactams (Figure 6).

Also, 89% (n = 17) of term neonates received antifungal treatment. The most prescribed group of antifungals was polyenes (AmB and LAmB) followed by the combination of AmB and FLC (Figure 7).

Angioinvasion was less frequent in this group, with gastrointestinal involvement observed in 8 cases and cutaneous infections in 2 cases.

Antifungal therapy was administered to 88.8% (n = 16) of term neonates, with a median treatment duration of 19.3 days. When used alone for rhino-cerebral infections (n = 2), the mortality rate was 50%. In contrast, no cases of gastrointestinal or cutaneous mucormycosis received antifungal therapy alone.

Surgical intervention was performed in 88.8% (n = 16) of the cases, with 2 neonates with gastrointestinal involvement undergoing surgery alone, resulting in a 100% mortality rate. Combined antifungal and surgical treatment was administered to 77.7% (n = 14) of neonates, yielding a mortality rate of 28.5% (n = 4).

Overall, the mortality rate among term neonates was 38.8% (n = 7), with 50% mortality in females (n = 2) and 38.4% in males (n = 5). By disease subtype, mortality rates were 50% (n = 1) for cutaneous mucormycosis, 35.7% (n = 5) for gastrointestinal mucormycosis, and 50% (n = 1) for rhino-cerebral mucormycosis.

The odds ratios for the associations between birth stage, sex, topography, antifungal treatment, surgical intervention, and pathogen are presented in Table 3, as determined through both univariate and multivariate analyses.

On the unadjusted analysis, extreme preterm status was associated with a significantly higher risk of mortality; 61.91% had an increased risk of death compared with preterm and very preterm neonates, who exhibited a mortality rate of 27.8% (OR = 7.273, 95% CI [1.301–36.39]; *p* = 0.014). Additionally, results of the univariate analysis revealed that neonates who did not receive antifungal treatment had an 85.71% probability of mortality, whereas those who received antifungal treatment had a mortality rate of 36.18% (OR = 10.59, 95% CI [2.406–50.02]; *p* = 0.002).

Moreover, a trend toward better outcomes was observed in patients with cutaneous mucormycosis, with a mortality rate of 31.58%, compared with a mortality rate of 53.84% in patients with gastrointestinal mucormycosis (OR = 0.396, 95% CI [0.1150–1.225]; *p* = 0.111). No other statistically significant differences were identified.

In the multivariate analysis, a trend was observed indicating that antifungal treatment may reduce mortality compared with untreated cases (OR = 11.25, 95% CI [0.01943–0.9974]; *p* = 0.068). However, this association did not reach statistical significance.

## 4. Discussion

This systematic review included 61 reported cases of neonatal mucormycosis worldwide [11,12,13,14,15,16,17,18,19,20,21,22,23,24,25,26,27,28,29,30,31,32,33,34,35,36,37,38,39,40,41,42,43,44,45,46,47,48,49,50,51,52,53,54], providing a significant source of information on a rare but highly lethal condition. This analysis documented epidemiological data, risk factors, clinical presentations, diagnostic methods, management, and outcomes.

Mucormycosis is characterized as a rare but lethal infection that primarily affects adults with some form of immunosuppression [3,4,55]. However, this fungal infection is also significant in a highly vulnerable population such as newborns, particularly those who are born preterm [5,18,37]. This state of vulnerability can be exacerbated by various risk factors. In this review, low birth weight, respiratory distress, and exposure to invasive therapies such as mechanical ventilation, catheter use, and broad-spectrum antibiotics were identified as key risk factors. Prematurity was present in 70.5% of the reviewed cases and was considered an immunocompromised state due to an undeveloped skin barrier, fragile intestinal mucosa, and an underdeveloped immune system [7,56]. Additionally, the prolonged use of antibiotics, observed in 67.2% of cases, induced intestinal dysbiosis, increasing susceptibility to opportunistic fungal infections such as mucormycosis [7,56]. This is consistent with studies such as that of Cotten et al., which demonstrated that prolonged neonatal exposure to antibiotics increases morbidity and the incidence of conditions such as necrotizing enterocolitis (NEC), one of the main differential diagnoses of a clinical variant of mucormycosis, intestinal mucormycosis [7,18,21,57].

Most cases were reported in developing countries, particularly in India, with 29 cases [15,16,17,18,19,20,21,22,23,24,44,45,46,47,48,49,50,51]. This can be attributed to a higher prevalence of risk conditions such as limited resources in neonatal intensive care units, unequal access to antifungal treatments, higher preterm birth rates [4,50,58,59,60], and the presence of concurrent diseases, such as the COVID-19 pandemic [4]. According to Ozbek et al., these findings highlight the need for specific strategies for the prevention and management of mucormycosis in neonates [4,5].

The review confirmed that *Rhizopus* and *Mucor* are the primary etiological agents of mucormycosis, consistent with previous studies that emphasize their ubiquity in the environment, particularly in soil and decomposing organic matter [55,59,60]. Among the analyzed cases, *Rhizopus* was the predominant species, reinforcing its clinical relevance as the dominant pathogen in this invasive fungal infection [55]. Additionally, an increase in its incidence was identified during the COVID-19 pandemic [4], reflecting changes in the epidemiology of these fungi due to the extensive use of steroids and antibiotics, as well as the immunosuppression associated with SARS-CoV-2 [4,55,56]. This shift also led to an increase in the rhino-orbito-cerebral variant in adults (15% to 32%) [4], while in pediatrics, Otto et al. reported 64 cases of rhinosinusal mucormycosis as the most common form [9]. This review found two cases of rhino-cerebral mucormycosis caused by *Rhizopus* spp. during the critical period of the COVID-19 pandemic [51,52]; however, this variant was much less prevalent in neonates, where the gastrointestinal variant was the most common, with 39 cases [11,12,14,17,21,24,25,27,31,34,35,36,44,45,46,47,48,50,54].

The gastrointestinal variant was the most common clinical presentation in neonates (63.9%), followed by cutaneous and rhino-orbital forms. Clinically, gastrointestinal mucormycosis manifested with abdominal distension (75%), bilious or fecaloid vomiting (41.6%), and nonspecific radiological signs such as fixed bowel loops and even intestinal perforation in advanced cases. The absence of intestinal pneumatosis, characteristic of bacterial NEC, was a key finding for clinical suspicion of mucormycosis [7,21]. The most affected segments were the colon and ileum, though less common locations such as the omental sac, jejunum, and rectum were also reported, suggesting that this pathology can affect any part of the gastrointestinal tract [4].

Gastrointestinal involvement was the most frequent presentation in both preterm and term neonates, possibly related to enteral feeding, aside from exclusive breastfeeding, intestinal perforations associated with low birth weight, and the angioinvasive nature of mucorales [47,61]. These neonates might be beneficiated from the use of some strains of probiotics [62], however; the high mortality rate (51.16% in preterm neonates) underscores the importance of early detection and timely surgical intervention.

The cutaneous form was more common in preterm neonates, with a mortality rate of 29.4% (n = 5), emphasizing the need for early intervention through antifungal therapy and surgical debridement. In many cases, its occurrence was associated with the use of dressings, catheters, and environmental factors. Its relationship with prenatal care and neonatal exposure to mucorales in the environment should also be discussed [63,64].

The diagnosis of mucormycosis in neonates was significantly delayed due to the nonspecific nature of the initial symptoms and their similarity to other clinical entities, particularly NEC in its gastrointestinal variant. In 98.3% of cases, the diagnosis was confirmed through histopathology or culture (Figure 8), highlighting the importance of early biopsies in high-risk patients and surgical intervention in gastrointestinal cases. The use of molecular biology techniques was not reported.

Although the optimal formulation of amphotericin B for mucormycosis is not yet fully determined, its intravenous administration at a concentration of 5 mg/kg/day showed good effectiveness. Combined with surgical debridement, this strategy proved to be the most effective, significantly improving survival rates. However, the overall mortality rate remains high (47.5%), particularly in the gastrointestinal presentation (53.8%). In this context, posaconazole can be considered an alternative to amphotericin B, although further clinical studies are needed to establish optimal treatment guidelines [65].

The high mortality associated with this condition is related to the strong angioinvasiveness of the fungus, which is linked to thrombocytopenia due to the response against the mucoral where platelets linked to spores and hyphae diminishing their growth. However, all the responses from innate and adaptative immunity produce an inflammation response strong enough to lead to tissue necrosis, emphasizing the aggressiveness of this fungal infection and the need for early intervention to mitigate its impact [66].

## 5. Limitations

Despite the thorough collection and analysis of neonatal mucormycosis cases, our study has several limitations. First, the lack of detailed data on disease duration in the analyzed cases prevents the construction of survival curves using the Kaplan–Meier method, limiting the ability to make precise comparisons regarding clinical progression and mortality.

Additionally, there is insufficient reporting on the time elapsed before initiating antifungal treatment, and in many cases, the exact doses administered are not specified. This gap makes it difficult to assess the impact of early treatment and the adequacy of therapeutic strategies used.

Another significant limitation is the absence of molecular information in most cases, preventing accurate identification of mucorales species. This restricts the ability to correlate specific species with clinical patterns and treatment responses.

Finally, the lack of more detailed reporting on clinical and epidemiological data in several of the included studies highlights the need to improve documentation, especially regarding gastrointestinal mucormycosis, the most frequently reported form. Considering the potential influence of enteral feeding type, future research should emphasize the benefits of breastfeeding and the possible role of probiotics in neonatal mucormycosis, particularly in its gastrointestinal presentation. Understanding the impact of feeding practices on intestinal vulnerability, as suggested by Hunter et al. in the context of necrotizing enterocolitis, can provide valuable insights into optimizing nutritional strategies for at-risk neonates [62].

## 6. Conclusions

Neonatal mucormycosis is a rare but highly fatal and invasive infection that primarily affects those with risk factors such as prematurity, especially in extreme prematurity cases. The gastrointestinal form is the most common and is often misdiagnosed as necrotizing enterocolitis, leading to delayed treatment. Although treatment with amphotericin B and surgical debridement appears to improve outcomes, the mortality rate remains alarmingly high. Additional research is crucial to refining treatment protocols and improving survival rates in affected neonates.

## Figures and Tables

**Figure 1 tropicalmed-10-00086-f001:**
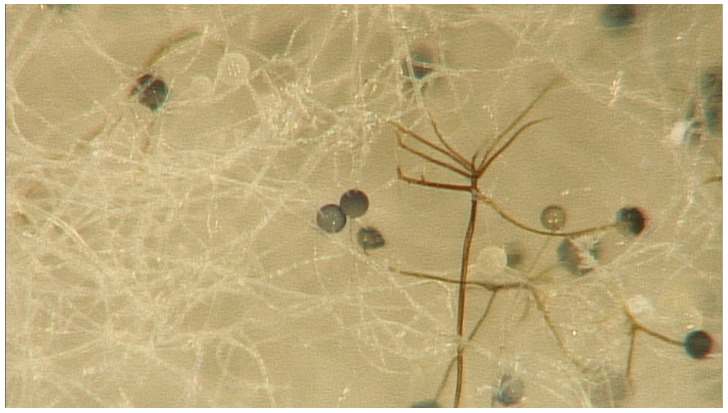
Microscopic study of *Rhizopus* spp. 40×, courtesy of Roberto Arenas.

**Figure 3 tropicalmed-10-00086-f003:**
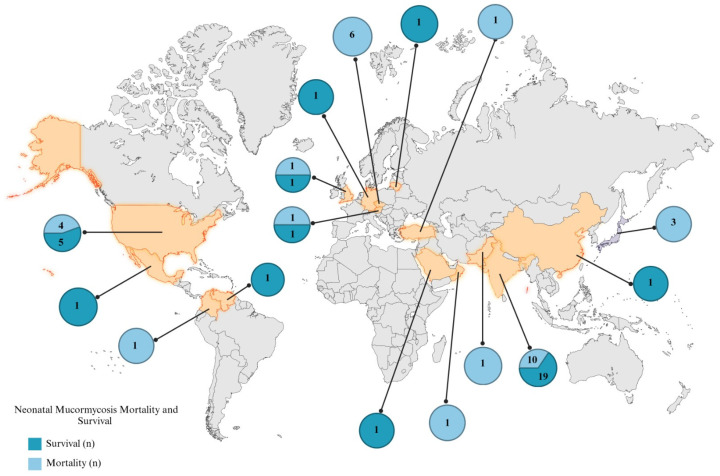
Global distribution. Neonatal Mucormycosis: Mortality and Survival. Dark blue are the cases reported of neonatal mucormycosis that survival. Light blue are the cases reported of neonatal mucormycosis that died.

**Figure 4 tropicalmed-10-00086-f004:**
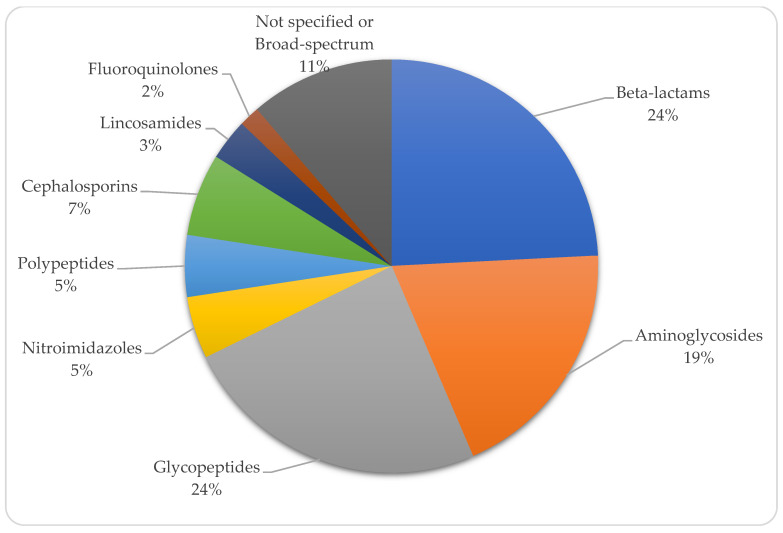
Distribution of antibiotic groups used in preterm neonates.

**Figure 5 tropicalmed-10-00086-f005:**
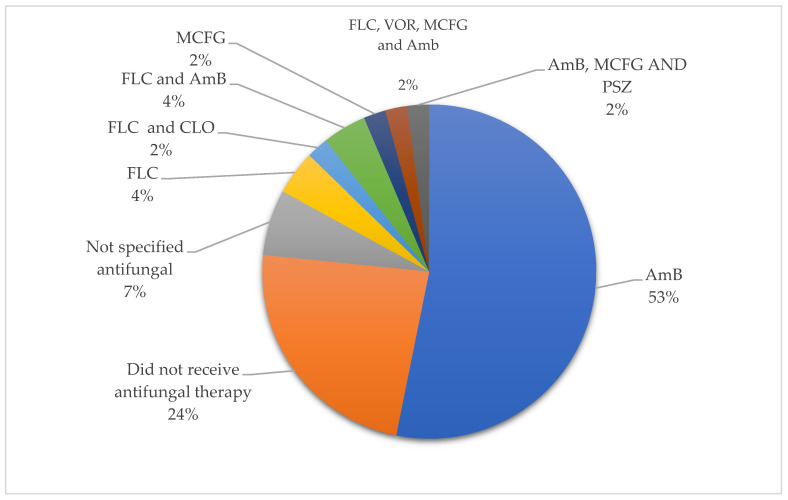
Distribution of antifungal used in preterm neonates.

**Figure 6 tropicalmed-10-00086-f006:**
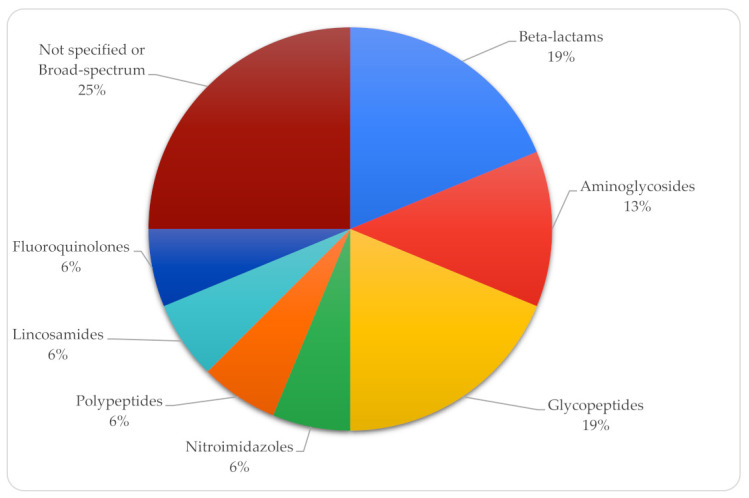
Distribution of antibiotic groups used in term neonates.

**Figure 7 tropicalmed-10-00086-f007:**
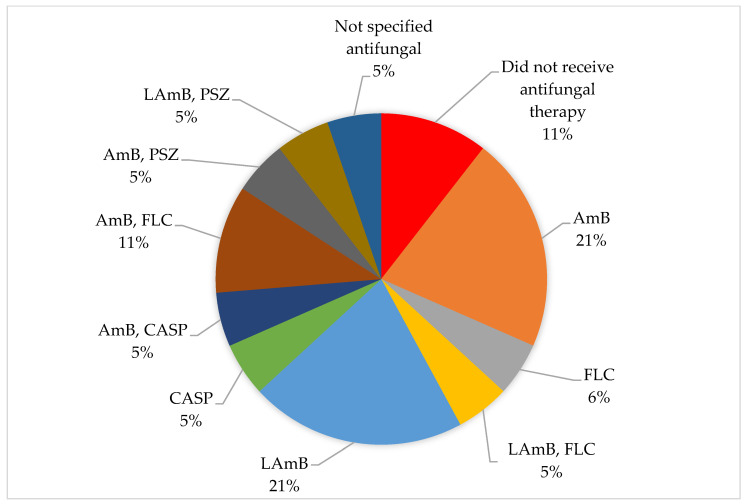
Distribution of antifungal used in term neonates.

**Figure 8 tropicalmed-10-00086-f008:**
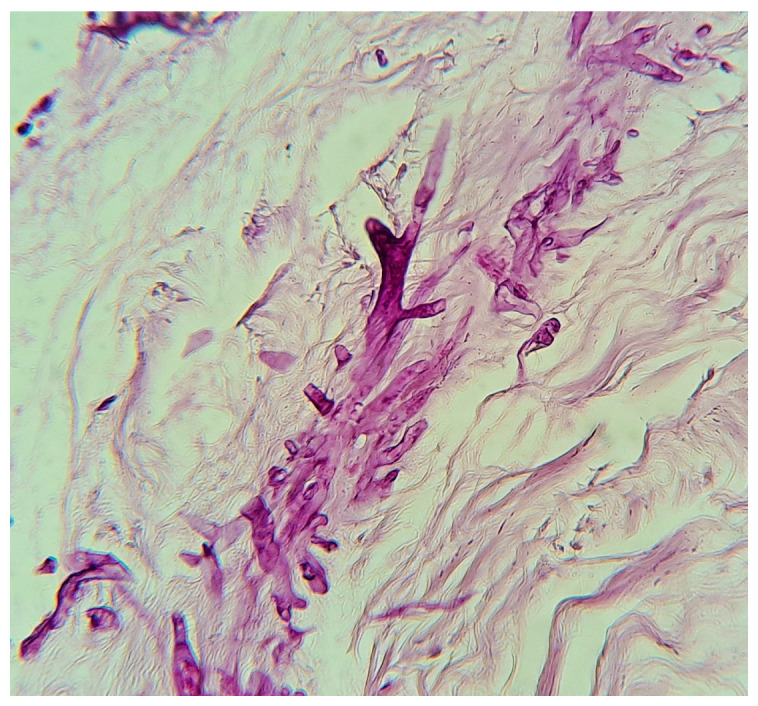
Biopsy of a mucormycosis case, PAS stain, 40×, courtesy of Roberto Arenas.

**Table 1 tropicalmed-10-00086-t001:** Cases of mucormycosis in preterm newborns.

Year	Country	Topography	Sex	Onset Age	Gestational Age (Weeks)	Birth Weight (kg)	Respiratory Distress	Oxygen	Diagnostic Method	Etiological Agent	Antibiotics	Surgical Treatment	Antifungal Therapy	Survival	Reference
2013	Austria	Gastrointestinal	M	21	36	NA	Yes	Yes	Biopsy, culture	*Mucor* spp.	Yes	Yes	AmB	Yes	[11]
2004	China	Gastrointestinal	M	15	26.4	0.839	Yes	Yes	Biopsy, GMS	*Mucor* spp.	Yes	Yes	AmB	Yes	[12]
2020	Colombia	Cutaneous	F	20	24.42	0.645	Yes	Yes	Biopsy, culture	*Rhizopus* spp.	Yes	No	AmB	No	[13]
2008	Czech Republic	Gastrointestinal	F	1	26	0.85	Yes	Yes	Biopsy	*Mucor* spp.	Yes	No	NO	No	[14]
2008		Gastrointestinal	NA	5	26	0.78	Yes	Yes	Biopsy	*Mucor* spp.	No	Yes	NO	No	
2008		Gastrointestinal	M	3	27	0.96	Yes	Yes	Biopsy	*Mucor* spp.	No	Yes	NO	No	
2008		Gastrointestinal	F	8	30	1.1	Yes	Yes	Biopsy	*Mucor* spp.	Yes	No	NO	No	
2008		Gastrointestinal	NA	6	NA	0.98	Yes	Yes	Biopsy	*Mucor* spp.	No	No	NO	No	
2008		Gastrointestinal	M	7	NA	1.9	Yes	Yes	Biopsy	*Mucor* spp.	No	Yes	NO	No	
2005	India	Cutaneous	M	10	35	1.6	Yes	Yes	Biopsy, culture	*Mucor* spp.	Yes	Yes	AmB	Yes	[15]
2006		Cutaneous	M	2	34	2	Yes	Yes	Biopsy	*Mucor* spp.	Yes	Yes	AmB	Yes	[16]
2007		Gastrointestinal	M	2	37	2.1	No	No	Biopsy, GMS	*Mucor* spp.	No	Yes	YES *	Yes	[17]
2012		Gastrointestinal	M	8	32	NA	No	No	Biopsy, GMS	*Mucor* spp.	No	Yes	AmB	No	[18]
2012		Gastrointestinal	F	22	32	NA	No	No	Biopsy, GMS	*Mucor* spp.	Yes	Yes	AmB	Yes	
2012		Gastrointestinal	F	7	33	NA	No	No	Biopsy, GMS	*Mucor* spp.	No	Yes	AmB	No	
2012		Gastrointestinal	F	10	33	1.6	No	No	Biopsy, GMS	*Mucor* spp.	Yes	Yes	AmB	Yes	
2012		Gastrointestinal	M	18	36	NA	No	No	Biopsy, GMS	*Mucor* spp.	Yes	Yes	AmB	Yes	
2013		Gastrointestinal	M	2	37	1.5	No	No	Biopsy	*Mucor* spp.	No	Yes	YES *	Yes	[19]
2016		Gastrointestinal	F	13	33	NA	Yes	Yes	Biopsy	*Mucor* spp.	Yes	Yes	NO	No	[20]
2018		Gastrointestinal	M	3	NA	NA	Yes	Yes	Biopsy	*Mucor* spp.	Yes	Yes	AmB	No	[21]
2018		Gastrointestinal	M	5	NA	NA	Yes	Yes	Biopsy	*Mucor* spp.	No	Yes	AmB	No	
2018		Cutaneous	M	5	25	0.5	Yes	Yes	Biopsy	*Mucor* spp.	Yes	Yes	AmB	No	[22]
2022		Cutaneous	M	11	30	1.8	Yes	Yes	Culture, KOH, biopsy	*Lichtheimia ramosa*	Yes	Yes	AmB	Yes	[23]
2022		Gastrointestinal	M	10	36	1.5	Yes	Yes	Biopsy	*Mucor* spp.	Yes	Yes	NO	Yes	[24]
2011	Japan	Gastrointestinal	M	1	26	0.59	Yes	Yes	Biopsy	*Mucor* spp.	Yes	Yes	FLC	No	[25]
2017		Disseminated infection	NA	7	22	0.452	No	No	Blood culture	*Mucor* spp.	No	No	MCFG	No	[26]
2017		Gastrointestinal	M	9	24.71	0.58	No	No	Biopsy, GMS	*Rhizopus* spp.	No	Yes	AmB	No	[27]
2014	Lithuania	Cutaneous	F	14	24	0.584	Yes	Yes	KOH, culture	*Syncephalastrum* spp.	Yes	No	FLC, CLO	Yes	[28]
2004	Mexico	Cutaneous	F	18	36	1.6	No	No	Culture, biopsy	*Absidia corymbifera*	Yes	Yes	AmB	Yes	[29]
2010	Saudi Arabia	Cutaneous	M	4	26	0.715	Yes	Yes	Biopsy	*Mucor* spp.	Yes	No	AmB	Yes	[30]
2011	Türkiye	Gastrointestinal	M	15	27	0.9	Yes	Yes	Biopsy	*Mucor* spp.	Yes	Yes	NO	No	[31]
2020	UK	Cutaneous	NA	6	23.28	0.521	No	No	Biopsy	*Rhizopus microsporus*	No	No	AmB	Yes	[32]
2022		Cutaneous	F	7	23.57	0.46	Yes	Yes	Biopsy	*Mucor* spp.	Yes	No	YES *	No	[33]
2004	USA	Gastrointestinal	F	15	32	2.94	Yes	Yes	GMS	*Mucor* spp.	Yes	Yes	NO	Yes	[34]
2004		Gastrointestinal	F	7	25	0.704	Yes	Yes	Biopsy, GMS	*Absidia corymbifera*	Yes	Yes	AmB	No	[35]
2014		Gastrointestinal	NA	7	29	1.4	No	No	Biopsy	*Mucor* spp.	No	Yes	NO	No	[36]
2017		Cutaneous	M	7	23.57	0.45	No	No	Biopsy, culture	*Rhizopus* spp.	Yes	No	AmB	Yes	[37]
2018		Cutaneous	M	6	24	NA	No	No	Culture	*Rhizopus* spp.	No	No	NO	No	[38]
2018		Cutaneous	F	6	25	0.806	Yes	Yes	Biopsy	*Mucor* spp.	Yes	Yes	FLC, VOR, MCFG, AmB	Yes	[39]
2019		Cutaneous	F	4	23.57	0.605	Yes	Yes	Culture, KOH, biopsy	*Rhizopus* spp.	Yes	Yes	FLC, AmB	No	[40]
2020		Cutaneous	M	14	24.57	0.585	Yes	No	Biopsy, culture	*Rhizopus* spp.	Yes	Yes	AmB, MCFG, PSZ	Yes	[41]
2022		Cutaneous	M	6	24.85	0.75	Yes	No	Culture, biopsy	*Rhizopus* spp.	Yes	No	AmB, FLC	Yes	[42]
2005	Venezuela	Cutaneous	F	5	34	2.7	Yes	Yes	Biopsy, KOH, culture	*Rhizopus* spp.	Yes	Yes	AmB	Yes	[43]

Yes *, not specified antifungal; GMS, Groccott’s AmB, Amphotericin B; FLC, Fluconazole; MCFG, Micafungine; PSZ, Posaconazole, CLO; Clotrimazole, VOR; Voriconazole.

**Table 2 tropicalmed-10-00086-t002:** Cases of mucormycosis in term newborns.

Year	Country	Topography	Sex	Onset Age	Gestational Age (Weeks)	Birth Weight (kg)	Respiratory Distress	Oxygen	Diagnostic Method	Etiological Agent	Antibiotics	Surgical Treatment	Antifungal Therapy	Survival	Reference
2013	Austria	Gastrointestinal	M	21	38	Na	Yes	Yes	Biopsy, culture	*Rhizopus*	No	Yes	NO	No	[11]
2009	Germany	Gastrointestinal	M	13	38	2.2	No	No	Biopsy	*Mucor* spp.	Yes	Yes	AmB	Yes	[44]
2004	India	Gastrointestinal	F	7	>38	Na	No	No	Biopsy	*Mucor* spp.	Na	Yes	FLC	Yes	[45]
2004		Gastrointestinal	M	16	>38	Na	Yes	Yes	Biopsy	*Mucor* spp.	Yes	Yes	NO	No	[45]
2004		Gastrointestinal	M	4	>38	Na	No	No	Biopsy	*Mucor* spp.	No	Yes	AmB	No	[45]
2010		Gastrointestinal	M	6	>38	2.75	Yes	Yes	Biopsy	*Mucor* spp.	Yes	Yes	LAmB, FLC	Yes	[46]
2011		Gastrointestinal	M	1	41	1.7	Na	No	Biopsy	*Mucor* spp.	Yes	Yes	LAmB	Yes	[47]
2012		Gastrointestinal	M	8	>38	Na	No	No	Biopsy, GMS	*Mucor* spp.	Yes	Yes	AmB	Yes	[18]
2015		Gastrointestinal	M	4	>38	1.75	No	No	Biopsy, GMS	*Mucor* spp.	No	Yes	AmB	Yes	[48]
2016		Cutaneous	NA	7	>38	2.3	Yes	Yes	Biopsy, GMS	*Mucor* spp.	No	Yes	LAmB, CASP	Yes	[49]
2018		Gastrointestinal	M	2	39	2.7	No	No	Biopsy	*Mucor* spp.	Yes	Yes	AmB, CASP	Yes	[50]
2018		Gastrointestinal	M	13	>38	Na	Yes	Yes	Biopsy	*Mucor* spp.	Yes	Yes	AmB, FLC	Yes	[21]
2018		Gastrointestinal	F	6	>38	Na	Yes	Yes	Biopsy	*Mucor* spp.	Yes	Yes	FLC, AmB	Yes	[21]
2018		Gastrointestinal	M	5	>38	Na	Yes	No	Biopsy	*Mucor* spp.	Yes	Yes	AmB, PSZ	No	[21]
2021		Rhino-cerebral	M	22	>38	2.2	Yes	Yes	KOH	*Rhizopus* spp.	No	No	LAmB, PSZ	Yes	[51]
2023		Rhino-cerebral	F	23	>38	2	Yes	Yes	Culture, KOH, biopsy	*Rhizopus* spp.	Yes	No	LAmB	No	[52]
2018	Oman	Cutaneous	F	5	>38	1.95	Yes	Yes	Biopsy, culture	*Mucor* spp.	Yes	Yes	LAmB	No	[53]
2019	Pakistan	Gastrointestinal	M	10	>38	Na	No	No	Biopsy	*Mucor* spp.	Yes	Yes	Yes *	No	[54]

Yes *, not specified antifungal; GMS, Groccott’s methanamine silver stain; LAmB, Liposomal Amphotericin B; AmB, Amphotericin B; CASP, Caspofungin; FLC, Fluconazole; PSZ, Posaconazole.

**Table 3 tropicalmed-10-00086-t003:** Univariate and multivariate analyses of the associations between birth stage, sex, topography, antifungal treatment, surgical intervention, and pathogen (OR, 95% Confidence Intervals, ***p***).

						Univariate Analysis			Multivariate Analysis		
Variables		Total N = 61	Mortality N = 29 (47.5%)	Survival N = 32 (52.5%)	Χ^2^	OR	CI 95%	*p*	OR	CI 95%	*p*
Birth stage Missing n = 4	Term	18 (100%)	7 (38.89%)	11 (61.11%)							
	Preterm	39 (100%)	18 (46.15%)	21 (53.85%)	0.264	1.347	0.4269 to 4.046	0.607	0.9835	0.1697 to 6.072	0.9851
	Preterm	12 (100%)	2 (16.67%)	10 (83.33%)							
	Very premature	6 (100%)	3 (50.00%)	3 (50.00%)							
	Preterm extreme	21 (100%)	13 (61.91%)	8 (38.09%)	6.064	7.273	1.301 to 36.39	0.014 *	0.436	0.06086 to 2.758	0.3851
Sex Missing n = 6	Female	19 (100%)	10 (52.64%)	9 (47.36%)							
	Male	36 (100%)	15 (41.67%)	21 (58.33%)	0.570	0.643	0.2188 to 2.025	0.571	0.506	0.1207 to 2.022	0.335
Topography Missing n = 1	Gastrointestinal	39 (100%)	21 (53.84%)	18 (46.16%)							
	Cutaneous	19 (100%)	6 (31.58%)	13 (68.42%)	2.546	0.396	0.1150 to 1.225	0.111	2.112	0.3590 to 16.61	0.432
	Rhino-cerebral	2 (100%)	1 (50.00%)	1 (50.00%)							
Antifungal Treatment	Yes	47 (100%)	17 (36.18%)	30 (63.82%)							
	No	14 (100%)	12 (85.71%)	2 (14.29%)	10.62	10.59	2.406 to 50.02	0.002 *	0.171	0.01943 to 0.9974	0.068
Surgical Intervention	No	14 (100%)	8 (57.14%)	6 (42.86%)							
	Yes	47 (100%)	21 (44.69%)	26 (55.31%)	0.672	1.651	0.5355 to 5.639	0.545	0.838	0.07576 to 8.367	0.880
Pathogen Missing n = 4	*Mucor* spp.	45 (100%)	22 (48.89%)	23 (51.11%)							
	*Rhizopus* spp.	12 (100%)	6 (50.00%)	6 (50.00%)	0.005	1.045	0.3081 to 3.536	>0.999	0.699	0.08821 to 4.827	0.375

* Statistically significant differences *p* ≤ 0.05.

## Data Availability

The original contributions presented in this study are included in the article. Further inquiries can be directed to the corresponding author.

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
