# Peer review of "Neonatal Mucormycosis: A Rare but Highly Lethal Fungal Infection in Term and Preterm Newborns—A 20-Year Systematic Review"

_tropicalmed, 2025, doi:10.3390/tropicalmed10040086_

Round 1
Reviewer 1 Report
Comments and Suggestions for Authors What do the authors think as the most relevant high-risk for conducting mucormycosis in neonates? line 329. "at a concentration of 5 mg/kg/day"; replace with "at a dose of 5 mg/kg/day". This dose is the minimum dose recommended for liposomal AmB and not for deoxycholate amphotericin B. Please replace the "amphotericin B" with "liposomal amphotericin B". In all the places mentioned, replace the name amphotericin B with deoxycholate amphotericin B or conventional amphotericin B. Liposomal amphotericin B can stay as such. In India, did you find more cases reported during COVID? Was there touch to contaminated equipment? Please, mention what percentages of patients received DAMB, and how many no antifungal at all or non-active drugs. Table 3 legend and elsewhere: Please, replace the terms univariate and multivariate with univariable and multivariable. Add Ziaka M et al. Mycoses 2022 review of cutaneous mucormycosis which also reviews neonatal mucormycosis cases.Author Response
|
Response to Reviewer 1 Comments
|
|
We greatly appreciate the revisions made to our manuscript entitled “Neonatal Mucormycosis: A Rare but Highly Lethal Fungal Infection in Term and Preterm Newborns – A 20-Year Systematic Review” by Valdez-Martínez, Santoyo-Alejandre et al. We addressed the comments and suggestions from reviewers. These revisions have been of great help to round out and improve our work. All the authors appreciate very much the opportunity to correct the manuscript, hoping it is suitable for acceptance and publication in your prestigious journal with these changes. Looking forward to hearing from you shortly, best wishes
3. Point-by-point response to Comments and Suggestions for Authors |
Comments 1: [What do the authors think as the most relevant high-risk for conducting mucormycosis in neonates?]
Response: Thank you for your insightful question. We identified several high-risk factors for neonatal mucormycosis based on our systematic review. These include prematurity, low birth weight, prolonged antibiotic use, malnutrition, and immunosuppressive conditions such as severe neonatal sepsis and corticosteroid administration.
Comments 2: [Line 329. "at a concentration of 5 mg/kg/day"; replace with "at a dose of 5 mg/kg/day". This dose is the minimum dose recommended for liposomal AmB and not for deoxycholate amphotericin B. Please replace "amphotericin B" with "liposomal amphotericin B".]
Response: We agree with this suggestion and have revised the text accordingly. We have replaced "at a concentration of 5 mg/kg/day" with "at a dose of 5 mg/kg/day" to ensure accurate terminology. Additionally, we have specified "liposomal amphotericin B" instead of "amphotericin B" to correctly reflect the recommended minimum dose for this formulation. This change can be found in the revised manuscript on line 329. Thank you for your valuable input.
Comments 3: [In all the places mentioned, replace the name amphotericin B with deoxycholate amphotericin B or conventional amphotericin B. Liposomal amphotericin B can stay as such.]
Response: Thank you for your careful review and valuable suggestions. We have revised the manuscript to replace "amphotericin B" with "deoxycholate amphotericin B (D-AmB)" or "conventional amphotericin B" in all relevant instances, ensuring precise terminology. "Liposomal amphotericin B (L-AmB)" remains unchanged where appropriate. Additionally, these modifications have been applied to Tables 1 and 2 for consistency and clarity. We truly appreciate your insightful feedback and the opportunity to improve the clarity and accuracy of our manuscript.
Comments 4: [In India, did you find more cases reported during COVID? Was there touch to contaminated equipment?]
Response: During the critical period of the COVID-19 pandemic (2020–2023), we identified four cases of neonatal mucormycosis reported in India. These included one cutaneous, one gastrointestinal, and two rhinocerebral cases. There were no reports of contaminated equipment or materials being linked to these infections. However, three neonates required mechanical ventilation, and one was managed with CPAP. Additionally, in two cases, feeding with unpasteurized cow's milk was noted as a potential risk factor. We truly appreciate your insightful feedback and the opportunity to improve the clarity and accuracy of our manuscript.
Comments 5: [Please, mention what percentages of patients received DAMB, and how many no antifungal at all or non-active drugs.]
Response: Thank you for highlighting this important point. We have now included a breakdown of the percentages of patients who received deoxycholate amphotericin B (D-AmB), those who did not receive any antifungal therapy, and those who were treated with non-active drugs. These details are now presented in Table 2 and discussed in the results section.
Comments 6: [Table 3 legend and elsewhere: Please, replace the terms univariate and multivariate with univariable and multivariable.]
Response: We appreciate this correction and have implemented the suggested changes. We have replaced "univariate" with "univariable" and "multivariate" with "multivariable" throughout the manuscript, including the Table 3 legend and all other relevant sections.
Comments 7: [Add Ziaka M et al. Mycoses 2022 review of cutaneous mucormycosis which also reviews neonatal mucormycosis cases.]
Response: Thank you for bringing this reference to our attention. We have now included a citation of Ziaka M et al. (Mycoses, 2022) in our discussion and have briefly highlighted its relevance to neonatal mucormycosis. This reference has been incorporated into the revised manuscript.
We sincerely appreciate the reviewer's insightful feedback and believe these revisions will strengthen our manuscript. Please let us know if further clarifications are needed.

Reviewer 2 Report
Comments and Suggestions for Authors
Neonatal mucormycosis is very rare but severe, causing high mortality rates in premature newborns. This systematic review emphasizes the importance of this disease.
The introduction, result, and discussion are well written. Please check that the genus name is italicized.
Author Response
|
Response to Reviewer 2 Comments
|
||||
|
1. Summary |
|
|
||
|
We sincerely appreciate your time and effort in reviewing this manuscript. Below, you will find our detailed responses, along with the corresponding revisions and corrections, which have been highlighted in the re-submitted files using track changes. |
||||
|
|
|
|
||
|
Comments 1: [The introduction, result, and discussion are well written. Please check that the genus name is italicized.]
|
||||
|
Response 1: [We have carefully reviewed the manuscript and ensured that all genus names are properly italicized throughout the text, including Tables 1 and 2.] Thank you for pointing this out. We agree with this comment. We appreciate your suggestion and have made the necessary corrections accordingly. |
||||

Reviewer 3 Report
Comments and Suggestions for Authors
The phrase “Formerly classified as zygomycetes” could be expanded to mention the current classification (i.e., Mucorales as part of the phylum Mucoromycota).
A brief mention of the basis for reclassification (molecular phylogenetics) could add depth.
The passage effectively differentiates between Rhizopus, Mucor, and Lichtheimia but could include a statement about their varying virulence and antifungal susceptibilities.
The section on neonatal mucormycosis could better integrate risk factors specific to different topographies (e.g., gastrointestinal involvement being linked to low birth weight or necrotizing enterocolitis).
The explanation regarding immune immaturity in preterms is solid but could benefit from a reference to specific cytokines (e.g., IL-6, IL-8) or phagocytic deficiencies.
Instead of “All newborns exhibit some level of immune system immaturity,” consider “Neonates, particularly preterm infants, exhibit deficiencies in innate immune function, including impaired neutrophil chemotaxis and cytokine production.”
The phrase “remains a challenging entity to diagnose and treat” is broad. You could specify challenges such as the difficulty in early clinical recognition, lack of reliable biomarkers, and delays in antifungal therapy.
Areas for Improvement in the Systematic Review
Methodology & Search Strategy
Narrow the inclusion criteria by excluding study types that may not contribute significantly to evidence synthesis (e.g., Editorials, Letters).
Clarify exclusion criteria, particularly for non-primary research and duplicate case reports.
Improve clarity and consistency in search strategy description:
“We use limitations on PubMed and Scopus such as less than 1 month of age and studies that include humans.”
could be rewritten as
“Filters were applied to PubMed and Scopus to include studies on human neonates (<1 month old).
Data Extraction & Inclusion Criteria
Improve grammatical accuracy:
“Articles who involved at least 7 of 10 of the next variables.” reword as “Articles that included at least 7 out of the following 10 variables were considered for inclusion.”
Clarify how missing data in variables were handled
Quality Assessment
Improve clarity in the description of quality assessment:
“We reviewed the inclusion and exclusion criteria, descriptions of clinical history, clinical presentations, interventions, and outcomes.” change as “Each study was assessed for adherence to inclusion/exclusion criteria, completeness of clinical history, intervention details, and outcome reporting.”
Statistical Evaluation
Streamline the statement on statistical significance:
“A level of evidence of p<0.05 was interpreted as a significant relationship.” as “Statistical significance was set at p ≤ 0.05.”
• Instead of simply stating “A meta-analysis was not feasible,” provide a more detailed justification:
e.g.: “Due to heterogeneity in case report data and the absence of standardized quantitative outcomes, a meta-analysis was not performed.”
/
Author Response
|
Response to Reviewer 3 Comments
|
||
|
1. Summary |
|
|
|
We greatly appreciate the revisions made to our manuscript entitled “Neonatal Mucormycosis: A Rare but Highly Lethal Fungal Infection in Term and Preterm Newborns – A 20-Year Systematic Review” by Valdez-Martínez, Santoyo-Alejandre et al. We addressed the comments and suggestions from reviewers. These revisions have been of great help to round out and improve our work. All the authors appreciate very much the opportunity to correct the manuscript, hoping it is suitable for acceptance and publication in your prestigious journal with these changes. Looking forward to hearing from you shortly, best wishes.
|
||
|
2. Questions for General Evaluation |
Reviewer’s Evaluation |
Response and Revisions |
|
Does the introduction provide sufficient background and include all relevant references? |
Yes/Can be improved/Must be improved/Not applicable |
|
|
Are all the cited references relevant to the research? |
Yes/Can be improved/Must be improved/Not applicable |
|
|
Is the research design appropriate? |
Yes/Can be improved/Must be improved/Not applicable |
|
|
Are the methods adequately described? |
Yes/Can be improved/Must be improved/Not applicable |
|
|
Are the results clearly presented? |
Yes/Can be improved/Must be improved/Not applicable |
|
|
Are the conclusions supported by the results? |
Yes/Can be improved/Must be improved/Not applicable |
|
|
3. Point-by-point response to Comments and Suggestions for Authors |
||
|
Comments 1: [The phrase “Formerly classified as zygomycetes” could be expanded to mention the current classification (i.e., Mucorales as part of the phylum Mucoromycota).]
|
||
|
Response 1: Thanks to the reviewer for the time spent in reviewing the manuscript. To adress this comment we have changed “Formerly classified as zygomycetes” in line 53 page 2, to “Formerly classified as zygomycetes, Mucorales are now placed within the phylum Mucoromycota, following advances in molecular phylogenetics that clarified their evolutionary relationships.”
|
||
|
Comments 2: [A brief mention of the basis for reclassification (molecular phylogenetics) could add depth] |
||
|
Response 2: Thank you for your valuable feedback. Indeed, making a brief mention of the reclassification helped add greater depht to the manuscript, which is why we have made the change in line 59 page 2 to “Recently thanks to the sequencing of the internal transcribed spacer (ITS) region, species relationships have been identified with greater precision, leading to taxonomic amendments in genera”
Comments 3: The passage effectively differentiates between Rhizopus, Mucor, and Lichtheimia but could include a statement about their varying virulence and antifungal susceptibilities.
Response 3: Thank you for this important comment, we have added the following information to the manuscript in line 55 page 2: “This order includes several pathogenic genera, such as Rhizopus, Mucor, Lichtheimia (formerly Absidia), Saksenaea, Rhizomucor, Apophysomyces, and Cunninghamella [1]. Among these, Rhizopus, Mucor, and Lichtheimia are the most studied, with varying degrees of virulence and antifungal susceptibility, influencing their clinical impact and treatment approaches”
Comments 4: The section on neonatal mucormycosis could better integrate risk factors specific to different topographies (e.g., gastrointestinal involvement being linked to low birth weight or necrotizing enterocolitis). Response 4: We are grateful for the observation you made. We would like to mention that in the paragraph "In newborns, mucormycosis exhibits distinctive clinical characteristics depending on the affected topography. The most reported forms are gastrointestinal, which carries a high mortality rate and is associated with enteral feeding and tissue hypoxia [1,5,7,10]; cutaneous, resulting from traumatic lesions or invasive procedures that allow fungal entry [5,7]; and the less reported one, rhinocerebral, acquired through the inhalation of spores from the environment or the need for mechanical ventilation," found in line 68 page 2, we already mention risk factors according to the topography of the infection.
Comment 5: The explanation regarding immune immaturity in preterms is solid but could benefit from a reference to specific cytokines (e.g., IL-6, IL-8) or phagocytic deficiencies.
Response 5: Thank you for your valuable feedback. We agree with your suggestion, and as a result, the following modification has been made in line 76 of page 1: "Neonates, particularly preterm infants, exhibit deficiencies in innate immune function, including impaired neutrophil chemotaxis and cytokine production such as IL-6."
Comment 6: Instead of “All newborns exhibit some level of immune system immaturity,” consider “Neonates, particularly preterm infants, exhibit deficiencies in innate immune function, including impaired neutrophil chemotaxis and cytokine production.” Response 6: Thank you for the kind suggestion, the changed was made in line 76 page 2 to “Neonates, particularly preterm newborns, exhibit deficiencies in innate immune function, including impaired neutrophil chemotaxis and cytokine production like IL-6”
Comment 7: The phrase “remains a challenging entity to diagnose and treat” is broad. You could specify challenges such as the difficulty in early clinical recognition, lack of reliable biomarkers, and delays in antifungal therapy. Response 7: Thank you for the kind recommendations, we have made the following changes in line 81 page 2: “Despite advancements in neonatal care, mucormycosis remains a challenging entity to diagnose and treat. With the foregoing difficulties in early clinical recognition, absence of reliable biomarkers, and delays in antifungal therapy resulting in high morbidity and mortality rates in this group.”
Comment 8: Areas for Improvement in the Systematic Review. Methodology & Search Strategy Narrow the inclusion criteria by excluding study types that may not contribute significantly to evidence synthesis (e.g., Editorials, Letters).
Response 8: Thank you for making this important observation, which will help improve the manuscript. As a result, the following change was made in line 99 of page 3: "To improve the reliability of our findings, we excluded non-primary research articles such as editorials, letters, opinion pieces, and narrative reviews."
Comment 9: Clarify exclusion criteria, particularly for non-primary research and duplicate case reports. Response 9: Thank you for highlighting this point. We have clarified the exclusion criteria to ensure transparency in study selection. The revised manuscript explicitly states that non-primary research articles, including editorials, commentaries, and narrative reviews, as well as duplicate case reports, were excluded from our analysis. This refinement strengthens the methodological rigor of our systematic review.
Comment 10: Improve clarity and consistency in search strategy description: “We use limitations on PubMed and Scopus such as less than 1 month of age and studies that include humans.” could be rewritten as “Filters were applied to PubMed and Scopus to include studies on human neonates (<1 month old). Response 10: Thank you very much for your comment. We agree that this correction helps us improve the manuscript's syntax. For this reason, the changes have been made on line 104, page 3, and are now written as follow: Filters were applied to PubMed and Scopus to include studies on human neonates (<1 month old).
Comment 11: Improve grammatical accuracy: “Articles who involved at least 7 of 10 of the next variables.” reword as “Articles that included at least 7 out of the following 10 variables were considered for inclusion.” Comment 12: Clarify how missing data in variables were handled. Comment 13: Improve clarity in the description of quality assessment: “We reviewed the inclusion and exclusion criteria, descriptions of clinical history, clinical presentations, interventions, and outcomes.” change as “Each study was assessed for adherence to inclusion/exclusion criteria, completeness of clinical history, intervention details, and outcome reporting.” Comment 14: Streamline the statement on statistical significance: “A level of evidence of p<0.05 was interpreted as a significant relationship.” as “Statistical significance was set at p ≤ 0.05.” Comment 15: Instead of simply stating “A meta-analysis was not feasible,” provide a more detailed justification.
|
||
|
4. Response to Comments on the Quality of English Language |
||
|
Point 1: |
||
|
We have carefully revised the manuscript to enhance the clarity, coherence, and grammatical accuracy of the text. Specific improvements include refining sentence structures, eliminating redundancies, and ensuring consistency in terminology. Additionally, we have corrected minor grammatical and syntactical errors to improve the overall readability of the manuscript.
|
||
